# Effect of a Multi-Strain Probiotic Supplement on Gastrointestinal Symptoms and Serum Biochemical Parameters of Long-Distance Runners: A Randomized Controlled Trial

**DOI:** 10.3390/ijerph19159363

**Published:** 2022-07-30

**Authors:** Joanna Smarkusz-Zarzecka, Lucyna Ostrowska, Joanna Leszczyńska, Urszula Cwalina

**Affiliations:** 1Department of Dietetics and Clinical Nutrition, Medical University of Bialystok, ul. Mieszka I 4B, 15-054 Bialystok, Poland; lucyna.ostrowska@umb.edu.pl (L.O.); joanna@zapolska.pl (J.L.); 2Department of Statistics and Medical Informatics, Medical University of Bialystok, ul. Szpitalna 37, 15-295 Bialystok, Poland; urszula.cwalina@umb.edu.pl

**Keywords:** diet, gastrointestinal symptoms, gastrointestinal disturbances, long-distance runners, probiotics, laboratory tests

## Abstract

As many as 70% of athletes who practice endurance sports report experiencing gastrointestinal (GI) symptoms, such as abdominal pain, intestinal gurgling or splashing (borborygmus), diarrhea or the presence of blood in the stool, that occur during or after intense physical exercise. The aim of the study was to evaluate the effect of a multi-strain probiotic on the incidence of gastrointestinal symptoms and selected biochemical parameters in the serum of long-distance runners. After a 3-month intervention with a multi-strain probiotic, a high percentage of runners reported subjective improvement in their general health. Moreover, a lower incidence of constipation was observed. In the group of women using the probiotic, a statistically significant (*p* = 0.035) increase in serum HDL cholesterol concentration and a favorable lower concentration of LDL cholesterol and triglycerides were observed. These changes were not observed in the group of men using the probiotic. Probiotic therapy may reduce the incidence and severity of selected gastrointestinal symptoms in long-distance runners and improve subjectively assessed health condition.

## 1. Introduction

Physical activity brings many health benefits to the human body. However, excessive exercise can produce adverse health effects. As many as 70% of athletes [1] who practice endurance sports report experiencing gastrointestinal (GI) disturbances, such as abdominal pain, intestinal gurgling or splashing (borborygmus), diarrhea or the presence of blood in the stool, during or after intense physical exercise [2]. The intensity of physical activity significantly affects the frequency of GI symptoms [3]. Recreational, low-intensity training several times a week can have a positive effect on intestinal peristalsis, preventing constipation, limiting the contact of pathogens with the intestinal mucosa, and thus reducing the risk of colorectal cancer [4]. The situation is different when it comes to low to moderate but long-lasting efforts. Activities such as a marathon or an ultramarathon may cause GI symptoms similar to those occurring in individuals practicing high-intensity sports. These complaints are typically an individual matter, but the percentage of runners who experience them is significant. This was confirmed in a study by Jeukendrup et al. among professional triathletes in which as many as 93% of the respondents reported GI symptoms during the competition, with two participants having to withdraw from the race because of severe vomiting and diarrhea [5].

Scientific research indicates several possible reasons for exercise-induced GI symptoms. One of the theories suggests insufficient blood supply to the GI tract (in particular, to the intestines) during training or competition, which is caused by the redistribution of blood to meet the increased oxygen demand of working muscles [6]. Exercise intensity at the level of 70% VO_2max_ may result in a 60–70% decrease in visceral flow. Studies show that ischemia caused by physical activity increases the frequency of GI symptoms even when blood flow is reduced by 50% [7]. Thus, long-distance running performed at a lower intensity (50–60% VO_2max_) may also contribute to GI symptoms.

Another hypothesis concerns dysbiosis and the toxic effect of pathogenic bacteria due to reduced local visceral blood flow and the translocation of pathogenic bacteria into the bloodstream [8]. This is a significant problem as it may affect up to 20–60% of competitors performing intense physical activity (training for around 4–6 h a day, 6 days a week, thus not allowing the body to regenerate) [9]. The intestinal barrier, which is essential for protecting the host against invading pathogens, also plays a crucial role in maintaining overall health [10]. The intestinal barrier can be adversely affected by great (≥60–70% VO_2max_) physical exertion (the circulatory–gastrointestinal pathway redistributes blood flow to working muscles and peripheral circulation, thus reducing total splanchnic perfusion, and the neuroendocrine pathway enhances sympathetic activation results in reduced the functional capacity of the GI system) and medication, in particular non-steroidal anti-inflammatory drugs (NSAIDs) and proton pump inhibitors (PPIs), as well as chronic stress (also associated with participation in sporting competitions) [10,11].

Probiotic therapy can eliminate GI symptoms, which has been confirmed in a study by Pugh et al. from 2019, conducted with a group of 24 runners, that aimed to determine the frequency of GI symptoms (flatulence, belching, nausea, vomiting and diarrhea). The participants used a probiotic containing 25 billion CFU of *Lactobacillus acidophilus* CUL60, *L. acidophilus* CUL21, *Bifidobacterium bifidum* CUL20 and *B. animalis* subsp. *lactis* CUL34 or placebo for 28 days before a marathon [12]. Symptoms were described in each of the four weeks of the experiment. The study demonstrated reduced prevalence of symptoms in the group taking the probiotic in the third and fourth weeks of the intervention compared to the first and second weeks of the study, while no differences were found in the placebo group. In addition, in the group of runners taking the probiotic, reduced severity of GI symptoms was observed during the marathon (mainly in its final stage).

Proper nutrition also plays a crucial role in preventing GI symptoms in sportspersons. It is believed that the causes of GI symptoms in sportspersons may be excessive consumption of carbohydrates before competition or training, excessive consumption of fats and proteins, excessive consumption of high-fiber foods, in particular those containing insoluble fiber, and hypertonic drinks [13]. Carbohydrates in the diet of physically active people should constitute at least 55% of the daily energy intake [14]. Protein is another essential nutrient because of its transport and regulatory functions and its involvement in the reconstruction of damaged muscle fibers through the synthesis of metabolic and contractile proteins [15]. Adequate fat consumption is also vital, especially the intake of mono- and polyunsaturated fatty acids which, as analyses show, are consumed in smaller amounts by athletes compared to saturated fatty acids or cholesterol [14]. During intense physical activity, runners may suffer from mineral deficiencies. Particular attention should be paid to the intake of iron [16], magnesium [17], calcium [18], sodium [19] and potassium [20].

The aim of the study was to evaluate the effect of a multi-strain probiotic on the incidence of GI symptoms and selected biochemical parameters in the serum of long-distance runners.

## 2. Materials and Methods

The study was approved by the Bioethics Committee of the Medical University of Bialystok, Poland, No. RI-002/81/2017 (ClinicalTrials.gov, accessed on 29 July 2022, Identifier: NCT04530929). Seventy individuals qualified for the randomized, double-blind trial. Inclusion criteria were: male/female, age range 20–60 years old, moderate (60% VO_2max_, 70% HR_max_, exercise HR range: 112–140 bpm/minute) or intense physical activity (80% VO_2max_, 85% HR_max_, HR range 136–170 bpm/minute) (long-distance running). Exclusion criteria were: diseases such as inflammatory bowel disease or food allergies, a pacemaker and current probiotic supplementation.

Study participants were long-distance runners, actively participating in distance running events with race distances > 100 km. The participants engaged in endurance training and long-distance running (>5 km per day; 5–7 days per week). In addition, the participants took part in strength training workouts lasting around 45 min each, 1–2 times a week. The mean weekly time spent running was 4.6 h for women and 8.4 h for men in the probiotic group, and 8.3 h for women and 6.7 h for men in the placebo group.

Individuals who met the eligibility criteria were assigned to either the probiotic or placebo group. A computer database was used to assign the runners to either group. Simple random assignment without replacement was used. Every second person randomly selected by the computer was assigned to the probiotic group (*n* = 35 people). The rest of the participants were assigned to the placebo group (*n* = 35 people). Randomly, the computer assigned 20 men and 15 women to the probiotic group, and 26 men and 9 women to the placebo group. The number of women enrolled in the study was far smaller than that of men, which is due to the fact that the total number of women running long distances is smaller than that of men. However, the authors focused on comparing men and women in the study to present the effects of probiotics on both genders. Two women (one from the probiotic group and one from the placebo group) withdrew from the study without providing a specific reason, while two women from the placebo group could not participate in the final stage of the study due to injury. All tests and analyses were completely anonymous and study participants gave their written informed consent for participation.

The intervention in the study was either a probiotic (SANPROBI BARRIER—produced by Sanprobi Ltd., Szczecin, Poland, commonly available in pharmacies in Poland), or a placebo (of identical appearance, size and taste, produced for the purpose of the present study). The probiotic used in the study contained the following bacterial strains: *B. lactis* W52, *Levilactobacillus brevis* W63, *L. casei* W56, *Lactococcus lactis* W19, *Lc. lactis* W58, *L. acidophilus* W37, *B. bifidum* W23, *Ligilactobacillus salivarius* W24 in a dose of 2.5 × 10^9^ CFU/g (1 capsule). The probiotic group received Capsule A and the placebo group received Capsule B. The participants were asked to start using either Capsule A or Capsule B after the completion of laboratory blood tests. They were asked to ingest 2 capsules of the product twice a day (morning and evening) for a period of 3 months. Each participant was provided with a sufficient supply of either probiotic or placebo capsules. 

Sixty-six people (34 people from the probiotic group (14 women and 20 men) and 32 people from the placebo group (6 women and 26 men)) participated in Stage II of the study. The mean age of women in the probiotic group was 37.21 ± 8.09 years, the mean body weight was 62.62 ± 5.65 kg and the mean height was 166.63 ± 3.81 cm, while the mean age of men was 40.85 ± 8.32 years, the mean body weight was 79.35 ± 7.11 kg and the mean height was 179.54 ± 4.25 cm. In the placebo group, the mean age of women was 33.33 ± 8.73 years, the mean body weight was 66.50 ± 9.26 kg and the mean height was 168.78 ± 3.11 cm, while the mean age of men was 38.61 ± 8.84, the mean weight was 80.98 ± 10.91 kg and the mean height was 179.70 ± 5.63 cm. The experimental design is presented in Figure 1.

### 2.1. Diet

At the beginning (Stage I) and the end of the study (Stage II), the participants described their usual meals on three consecutive days, including two working days and one day off. Food diaries included meal timings, weight of products and dishes consumed, and fluid intake during the day. The participants were asked to follow their usual diet without any significant modifications throughout the duration of the study. Portion sizes were verified based on the “Album of photographs of food products and dishes” (published by the Polish Institute of Food and Nutrition, Warsaw, Poland) [21]. In the evaluation of the three-day dietary diaries provided by the participants, daily energy intake, content of the main nutrients, vitamins, minerals, cholesterol and dietary fiber were taken into consideration, and the results were compared with the findings of studies on long-distance running [14,15,16,17,18,19,20,21,22,23,24,25,26,27,28,29]. Data obtained from daily food rations were analyzed using a computer program “Dieta 5” (developed by the National Institute of Public Health, Warsaw, Poland).

### 2.2. Gastrointestinal Symptoms

At the beginning (Stage I) and end of the study (Stage II), the participants were asked to complete original questionnaires that aimed to determine the frequency and severity of GI symptoms before, during and after physical activity. The questionnaire contained single and multiple-choice questions.

### 2.3. Blood Tests

Blood tests were performed following the collection of 10 mL of venous blood from the ulnar vein. Two sets of tests (at the beginning and end of the study) were performed: complete blood count with blood smear, fasting glucose, total cholesterol, HDL cholesterol, LDL cholesterol and triglycerides, sodium, iron, potassium, magnesium and calcium in serum. For the biochemical tests, the blood was centrifuged, and determinations were made after the clot was separated in the blood serum.

### 2.4. Statistical Analysis

The statistical analysis was performed using STATISTICA 13.3 (StatSoft, Cracow, Poland). Descriptive statistics were developed by designating mean values, standard deviations, standard error, 95% CI, ranges of minimum and maximum values and medians for quantitative features, and numbers and percentages for qualitative features. Due to the size of the studied groups, the consistency of the distribution of the analyzed quantitative variables with the normal distribution was not assessed. Non-parametric methods were used in the analysis. For the dependent samples, the Wilcoxon signed rank test was used, whereas the Mann–Whitney U test was used for the independent samples. The analysis of the qualitative data was performed using the Pearson chi-square test and the exact Fisher test. Statistically significant results were set at the level of *p* < 0.05.

## 3. Results

The frequency and severity of the GI symptoms experienced by the participants were assessed at the beginning and the end of the study. The first of the analyzed symptoms was regurgitation of gastric contents into the esophagus during competition or training. At the beginning of the study, this symptom was reported by 43% of probiotic women (*n* = 6) and 50% of placebo women (*n* = 3) (*p* = 1.000) as well as 45% of probiotic men (*n* = 9) and 31% of placebo men (*n* = 3) (*p* = 0.368). At 3 months after the initiation of the intervention, in the group of runners taking the probiotic, symptom reduction was observed in 57% of women and 45% of men. In the placebo group, 50% of women and 46% of men reported a reduction in symptoms. The differences between the groups were not statistically significant. None of the participants reported a regurgitation of gastric contents into the esophagus during competition or training.

The second symptom investigated in the present study was the incidence of diarrhea. The majority of study participants reported experiencing diarrhea during the initial assessment: 79% of probiotic women (*n* = 11) and 67% of placebo women (*n* = 4) (*p* = 0.612), as well as 60% of probiotic men (*n* = 12) and 81% of placebo men (*n* = 21) (*p* = 0.187). The analysis of results regarding the incidence of diarrhea at 3 months after the initiation of the intervention showed a reduction in symptoms in the majority of respondents: 57% of probiotic women (*n* = 6) and 67% of placebo women (*n* = 3) (*p* = 0.199), as well as in 40% of probiotic men (*n* = 5) and 65% of placebo men (*n* = 14) (*p* = 0.225).

The third symptom analyzed in the study was constipation. During the initial assessment, it was found that constipation was reported more often by women from both groups in comparison to men (probiotic women (64%, *n* = 9) and placebo women (67%, *n* = 4); *p* = 1.000). A decreased incidence of constipation was reported by men from the probiotic group (11%, *n* = 3) and the placebo group (25%, *n* = 5) (*p* = 0.267). At 3 months after the initiation of the intervention, a reduction in the incidence of constipation was found, mainly among runners taking the probiotic (57% of women (*n* = 5) and 40% of men (*n* = 1)). In the placebo group, a reduction in the incidence of constipation was also shown (33% of women (*n* = 1) and 27% of men (*n* = 1)).

Alternating episodes of constipation and diarrhea are symptoms that may suggest irritable bowel syndrome (IBS). During the initial assessment, it was found that these disturbances were reported more often by women from both groups (probiotic = 43%, *n* = 6; placebo = 50%, *n* = 3; *p* = 1.000), and they were reported less frequently by men from the probiotic group (25%, *n* = 5) and the placebo group (8%, *n* = 2) (*p* = 0.212). At the final assessment, three months after the initiation of the intervention, only 7% of probiotic women (*n* = 1), 10% of probiotic men (*n* = 1) and 12% placebo men (*n* = 1) were still reporting alternating bouts of constipation and diarrhea. The majority of the respondents did not report the above symptoms.

In the final stage of the study, the participants were asked to provide a subjective evaluation of their overall health after a 3-month probiotic/placebo intervention. A far higher percentage of women who took the probiotic (71%, *n* = 10) reported an improvement in their health compared to 50% of women from the placebo group (*n* = 3) (*p* = 0.612). Furthermore, a larger percentage of probiotic men (60%, *n* = 12) reported an improvement in their health compared to men in the placebo group (50%, *n* = 13) (*p* = 0.351). One man from the probiotic group stated that his health had deteriorated. A larger percentage of placebo women (50%, *n* = 3) observed no change in their health in comparison to the probiotic group (29%, *n* = 4), although these differences were not statistically significant. Similar results were obtained for male participants, where 50% of men from the placebo group (*n* = 13) reported no change in their health in comparison to 35% of men from the probiotic group (*n* = 7). The differences between the groups were not statistically significant.

Serum biochemical parameters were also assessed at the beginning and the end of the study, and the results are shown in Table 1 and Table 2. At 3 months after the initiation of the intervention, the concentrations of sodium, potassium and iron in the sera of probiotic women increased, while the concentrations of iron decreased significantly in the placebo group. Moreover, in both studied groups of women, a decrease in fasting serum glucose was demonstrated. In the probiotic group, a statistically significant (*p* = 0.035), favorable increase in HDL cholesterol concentration was demonstrated, which was not observed in the placebo group, where the concentration of HDL cholesterol decreased. An improvement in the lipid profile in terms of LDL cholesterol and triglycerides was also shown in probiotic women. On the other hand, unfavorable changes were noted in the placebo group of women, whose LDL cholesterol concentration increased.

At 3 months after the initiation of the intervention, in both studied groups of men, sodium concentration increased, which in the placebo group was close to statistical significance (*p* = 0.061). In probiotic men, an increase in potassium concentration was observed, while in the placebo group a decrease was noted. In both studied groups, a decrease in serum calcium concentration was demonstrated, but a statistically significant result was observed only in the probiotic group (*p* = 0.011). The concentration of magnesium decreased in the probiotic group and increased in the placebo group (*p* = 0.034). Iron concentration increased in the placebo group and decreased in the probiotic group. At 3 months after the initiation of the intervention, the concentration of HDL cholesterol in both studied groups decreased. In probiotic men, an increase in total cholesterol and LDL fraction (*p* = 0.038) as well as triglyceride concentration was noted. In the placebo group of men, an increase in the concentration of LDL cholesterol was noted, as well as a decrease in the concentration of total cholesterol and triglycerides. In probiotic men, a favorable decrease in fasting glucose concentration was demonstrated, which was not shown in the placebo group, where the concentration of fasting glucose increased.

Selected parameters of the staple diet of the participants from the studied groups, divided by gender, were assessed at the beginning and the end of the study. The data are presented in Table 3. The initial assessment revealed that the mean daily energy intake was too low in both the probiotic and placebo women, and the differences between the groups were statistically significant (*p* = 0.018). Furthermore, the mean daily energy intake was also too low in both the probiotic and placebo men. It was also demonstrated that the total daily protein intake was too low in relation to the demand (about 80–90 g) in both studied groups of women. Similarly, in the probiotic group of men, the mean daily protein intake was too low in relation to the demand (around 100 g), while in the placebo group of men the dietary allowance for protein was met (105 g/day). The higher total daily protein intake in the group of placebo men compared to probiotic men was close to statistical significance (*p* = 0.055). Additionally, the placebo group of men showed a significantly higher (*p* = 0.024) mean consumption of animal protein compared to the probiotic group.

Next, total carbohydrate consumption was assessed. It was demonstrated that in the probiotic group of women the mean carbohydrate intake was 225.26 ± 13.77 g/day, while in the placebo group it was 173.13 ± 10.68 g/day (*p* = 0.028). Probiotic women did not meet their dietary allowance for carbohydrates (the norm is approximately 300 g/day), and neither did placebo women (approximately 330 g/day). Similarly, men from the probiotic and placebo groups did not meet their dietary allowance for carbohydrates (intake in these groups should be around 400 g/day). Dietary fiber consumption in both groups of women was below the recommended daily intake (25–27 g/day). In both studied groups of men, the consumption of dietary fiber was marginally lower than the recommended daily intake.

Total fat consumption was too low and cholesterol intake was too high in both studied groups of women. In the placebo group of men, cholesterol consumption was significantly higher (*p* = 0.023) than in the probiotic group. In both groups of men, the consumption of dietary cholesterol exceeded the recommended norms. It was demonstrated that the diet of both probiotic and placebo groups (of both sexes) should contain more monounsaturated and polyunsaturated fatty acids.

The daily intake of minerals by men and women from the studied groups was also assessed. A very high intake of sodium was noted, higher in the probiotic group of women, and the differences were statistically significant (*p* = 0.043). Women from both groups did not meet their dietary allowance for potassium, calcium or iron. However, a higher intake of potassium with food was reported by women from the probiotic group in comparison to those from the placebo group. The intake of calcium in the probiotic group was also higher and the differences were close to statistical significance (*p* = 0.063). Probiotic women met their dietary allowance for magnesium, in contrast to placebo women whose dietary allowance for this mineral was not met. Men from both groups met their dietary allowance for iron but exceeded their dietary allowance for sodium. Men from both groups did not meet their dietary allowance for potassium, calcium or magnesium.

Assessment of the study participants’ diet at 3 months after the initiation of the intervention did not reveal any statistically significant differences in the intake of nutrients between the probiotic and the placebo groups, which indicates that diet did not have an impact on the GI symptoms in the examined athletes.

## 4. Discussion

Regurgitation of gastric contents into the esophagus and, in some cases, vomiting during very intense exercise may exert a detrimental impact on the quality of training and performance during competition. The symptom was assessed in all studied groups, and it was demonstrated that a 3-month intervention with a probiotic/placebo produced a statistically insignificant decrease in incidence in the groups using the probiotic compared to the placebo. A recent study by Cheng et al. published in 2020, which is a meta-analysis of 14 clinical trials investigating the role of probiotic therapy in reducing GERD symptoms such as heartburn and regurgitation of gastric contents into the esophagus, revealed that probiotics can accelerate gastric emptying by interacting with mucosa receptors [30]. Moreover, probiotics preventing dysbiosis in the small intestine and the development of small intestinal bacterial overgrowth (SIBO) have a positive effect on GERD symptoms. However, the authors of the study emphasized that only 5 out of 14 publications were of good quality. Therefore, the authors highlight the need to design randomized placebo-controlled trials, with more participants and a longer duration of probiotic intervention, to fully confirm the beneficial effect of probiotics on GERD symptoms.

Participation in competitions and intensive training may also be hindered by diarrhea [31]. The microbiome of athletes is significantly different compared to healthy people who do not exercise [32]. The incidence of diarrhea in physically active people, apart from changes in the microbiome and transient dysbiosis caused by intense exercise, as well as taking medication (NSAIDs, PPIs, antibiotics), is influenced by diet (high protein, low carbohydrate) and supplementation (protein supplements) [33]. In our study, diarrhea was reported by the vast majority of runners. Study participants associated diarrhea with the intensity and duration of training rather than participation in competitions. At 3 months after the initiation of the probiotic/placebo intervention, a reduction in the incidence of diarrhea was demonstrated in a significant number of people in the placebo groups compared to the probiotic groups. The incidence of diarrhea may be influenced by the excessive consumption of hypertonic fluids and dietary fiber intake. The majority of runners reported a daily consumption of sports drinks. The intake of such drinks was declared by 93% of probiotic women (*n* = 13) and 83% of placebo women (*n* = 5), and 90% of probiotic men (*n* = 18) and 73% of placebo men (*n* = 19). The inter-group differences were not statistically significant. Stress during intense periods of training and competitions may also have an impact on the incidence of diarrhea. It may also be influenced by individual differences in the gut microbiome.

Constipation causes great discomfort during training and competition. Athletes can prevent constipation by increasing their intake of dietary fiber, but they must remember that excessive consumption of fiber may cause diarrhea. The assessment conducted at the beginning of the study revealed that constipation was more common among women from both the probiotic and placebo groups. At 3 months after the initiation of the intervention, a decreased incidence of constipation was reported by the participants, mainly by those taking the probiotic. Constipation could have been caused by too low an intake of dietary fiber in all studied groups. The insufficient consumption of fiber may have been associated with restrictions on diet modification set at the beginning of the study and the participants’ concerns regarding diarrhea.

The beneficial effect of probiotic strains on the GI symptoms experienced by athletes during intense and long-lasting sporting events has been demonstrated by a number of authors. By way of illustration, it has been revealed that supplementation with the probiotic strain *Limosilactobacillus fermentum* VRI-003 PCC reduced the frequency, duration and severity of GI symptoms on a 3-point scale [1]. Another study conducted in a group of professional cyclists and triathletes showed that supplementation with the *Ls. fermentum* PCC^®^ strain at a dose of 1 × 10^9^ CFU had a positive effect on reducing the incidence of GI symptoms, which appeared mainly during intense, strenuous workouts [34].

Studies on the influence of probiotic strains on the incidence of GI symptoms are not consistent. An example is a study in which supplementation with *B. animalis* subsp. *lactis* B1-04 (B1-04) 2.0 × 10^9^ CFU or *L. acidophilus* NCFM and *B. animalis* subsp. *lactis* B1-07 5 × 10^9^ CFU did not have a significant impact on the incidence of GI symptoms [35]. Furthermore, the group taking the B1-04 strain showed decreased exercise capacity than the NCFM and B1-07 group compared to the placebo group. Another example is a study in which 141 marathon runners were administered the probiotic strain *Limosilactobacillus rhamnosus* GG (LGG) or a placebo for 3 months [36]. The study did not find any differences in the incidence of both upper respiratory tract infections and GI disturbances between the group of athletes receiving the probiotic and the group taking the placebo.

The most recent review of the available literature on the effect of probiotic strains on the incidence of GI symptoms indicates that supplementation with probiotics has a beneficial effect on GI symptoms. However, there is a dearth of research on the use of probiotics in sport and, therefore, a change in methodology is needed so that specific probiotic strains for specific intensities of physical activity are identified [37].

Intense physical activity may have an impact on the concentration of some biochemical parameters in the serum of athletes. An increase in plasma volume, erythrocyte count and serum hemoglobin levels is then observed. It also has a very positive effect on the lipid profile, reducing the concentration of total cholesterol and LDL cholesterol and increasing the concentration of HDL cholesterol [38].

At 3 months after the initiation of the intervention, no statistically significant differences in the concentration of the morphological parameters of blood serum were observed in either the probiotic or placebo groups. In probiotic men and women, a decreased concentration of white blood cells, red blood cells, hemoglobin and platelets was observed in comparison to the initial blood test results. Similar results were reported in a study by Huang et al. [39]. A 6-week intervention with the probiotic strain *Lactiplantibacillus plantarum* TWK10 resulted in a decreased concentration of white blood cells in the study participants. However, in contrast to our findings, the study demonstrated an increased concentration of red blood cells in the participants. A decreased platelet count was consistent with the results of our study. As in the analysis of our results, no statistical significance was demonstrated for these parameters.

In the present study, the analysis of serum for mineral content revealed increased concentrations of sodium, potassium and iron in the sera of probiotic women at 3 months after the initiation of the intervention. The differences were not statistically significant. Importantly, in this group of women, iron concentration increased, while a significant decrease was observed in the concentration of this parameter in the placebo group (136.00 ± 21.01 µg/dL vs. 94.66 ± 14.12 µg/dL). Similar results were obtained in a study by Hoppe et al. in which increased iron absorption was demonstrated in young, healthy menstruating women after administration of the probiotic *Lp. plantarum* 299 v strain [40]. The women were not very physically active in comparison to the women in our study, but the above study may help to explain why serum iron levels in probiotic women investigated in the present study increased when the diet did not change. Optimal iron levels are important for improving maximal aerobic capacity in physically active people. Low iron levels have been shown to adversely affect performance due to impaired muscle function, oxidative metabolism and decreased physical efficiency [41]. Moreover, iron loss in physically active women is greater than in inactive women. The presence of heavy menses is also important. As women from the probiotic and control groups did not meet their dietary allowance for iron, their diets should be enriched with products containing heme iron as studies have shown that iron loss in endurance athletes is 30% to 70% greater due to physical activity [42].

In our research, a statistically significant decrease in calcium concentration was found in the group of probiotic men (*p* = 0.011) compared to the placebo group. In the group of women, both probiotic and placebo, no changes in calcium levels were found after 3 months of the study. Similar results to our research were obtained in a study in which 12 ultramarathon runners were analyzed [43]. The regulation of serum calcium levels is controlled by parathyroid hormone calcitonin and vitamin D_3_, underlining the importance of optimal intake of vitamin D_3_ with the diet. Barrack et al. [44] found that 85% of runners with increased bone turnover do not meet the AI standard for calcium. Our research was carried out in the autumn, which may affect the low exposure to sunlight and vitamin D_3_ synthesis.

At 3 months after the initiation of the intervention, a significant, favorable increase in HDL cholesterol concentration (*p* = 0.035) was demonstrated in probiotic women in contrast to placebo women, in whom HDL cholesterol levels decreased (the difference was not statistically significant). An improvement in the lipid profile in terms of LDL cholesterol and triglyceride levels was also shown in probiotic women. On the other hand, unfavorable changes occurred in placebo women, in whom the concentration of LDL cholesterol increased. Similar results were obtained in a study by Szulińska et al. [45] in which the same probiotic was used and, similar to our research, the concentration of HDL cholesterol in the group using the probiotic increased, but the differences were not statistically significant (*p* = 0.863). As in the analysis of our results, a statistically significant reduction in LDL cholesterol concentration was demonstrated in the group using the probiotic (*p* = 0.016). Similar to our results, the study revealed decreased triglyceride levels in both probiotic and placebo groups, although no statistical significance was shown for the observed differences.

As demonstrated by the results of our research and the findings of previously published studies on the impact of probiotic strains on athletes, more randomized trials are needed to determine the mechanism of the changes. Undoubtedly, probiotics have a beneficial effect on GI symptoms, and this may translate to improved athlete performance. The more intense the training, the better the preparation for competitions and, consequently, improved results are obtained by athletes. In order to elucidate the mechanism of action of probiotics on the bodies of physically active people, it would be necessary to analyze the intestinal microbiome of a particular athlete and, based on the findings, offer sportsperson-tailored probiotics therapy. 

The study has some limitations. One of them is a small sample size (*n* = 66) due to the high cost of the analyses. It is known that studies with larger sample sizes are more representative and more reliable. Furthermore, as there are more male than female runners, the genders may not be equally represented. Additionally, a study of a longer duration may have allowed for more precise conclusions to be drawn. Another factor limiting our study is the diversity in the microbiome of study participants, which would require the host-specific selection of probiotic strains. Research investigating the most appropriate probiotic strains for reducing GI symptoms in athletes is still ongoing. Furthermore, the same probiotic was given to all participants from the probiotic groups during the trial, although each individual reported different GI problems at the beginning of the study. 

## 5. Conclusions

In the group of long-distance runners taking the multi-strain probiotic, a higher percentage of participants reported an improvement in general health and a decreased incidence of constipation following a 3-month intervention.After 3 months of intervention in the group of women using the probiotic, a statistically significant increase in serum HDL cholesterol and a decrease in LDL cholesterol and triglycerides were observed. This was not observed in the group of men using the probiotic.The diet of long-distance runners should be more balanced, as it was demonstrated that the dietary intake of calories, nutrients, minerals and vitamins was inadequate.

## Figures and Tables

**Figure 1 ijerph-19-09363-f001:**
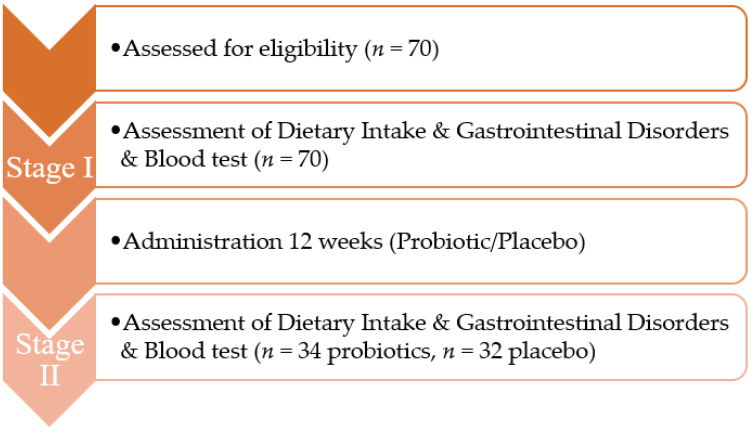
Seventy individuals met the study inclusion criteria. Stage I of the study included assessment of dietary intake, GI symptoms and blood tests. For 3 months following Stage I, participants took either a probiotic (*n* = 35) or a placebo (*n* = 35). After this time, Stage II of the study was conducted in which tests analogous to those in Stage I were performed in 34 people from the probiotic group and 32 people from the placebo group.

**Table 1 ijerph-19-09363-t001:** Comparison of blood biochemical parameters in women from the probiotic and placebo groups at the initial and final stages of the study. * Statistical significance (*p* < 0.05) determined between the initial and final stages.

Probiotic Women (*n* = 14)						Placebo Women (*n* = 6)				
	Initial Stage		Final Stage			Initial Stage		Final Stage		
Parameter	Mean ± SEM	95% CI	Mean ± SEM	95% CI	*p*	Mean ± SEM	95% CI	Mean ± SEM	95% CI	*p*
**WBC (×10^3^/μL)**	5.93 ± 0.40	5.06–6.81	5.84 ± 0.39	4.99–6.70	0.875	6.19 ± 0.43	5.08–7.31	5.99 ± 0.49	4.71–7.26	0.753
**RBC (×10^6^/μL)**	4.59 ± 0.07	4.42–4.75	4.55 ± 0.08	4.37–4.74	0.550	4.45 ± 0.08	4.23–4.66	4.49 ± 0.11	4.20–4.78	0.753
**HGB (g/dL)**	13.39 ± 0.18	12.98–13.80	13.20 ± 0.18	12.80–13.61	0.432	13.71 ± 0.18	13.25–14.18	13.68 ± 0.16	13.27–14.09	0.833
**PLT (×10^3^/μL)**	262.35 ± 19.86	219.44–305.27	251.85 ± 18.19	212.54–291.16	0.470	256.33 ± 23.88	194.93–317.72	265.83 ± 21.29	211.09–320.57	0.463
**Sodium (mmol/L)**	136.78 ± 0.29	136.13–137.43	137.42 ± 1.06	135.13–139.72	0.328	138.33 ± 0.98	135.79–140.87	138.16 ± 0.54	136.77–139.56	0.345
**Potassium (mmol/L)**	4.58 ± 0.10	4.35–4.81	4.65 ± 0.13	4.36–4.95	0.572	4.77 ± 0.11	4.46–5.07	4.95 ± 0.14	4.57–5.33	0.248
**Calcium (mmol/L)**	2.41 ± 0.02	2.35–2.46	2.41 ± 0.03	2.34–2.48	0.900	2.48 ± 0.04	2.36–2.61	2.40 ± 0.03	2.32–2.48	0.208
**Magnesium (mmol/L)**	0.83 ± 0.01	0.79–0.87	0.83 ± 0.02	0.77–0.89	0.900	0.84 ± 0.02	0.77–0.91	0.83 ± 0.01	0.80–0.86	0.529
**Iron (µg/dL)**	102.78 ± 12.71	75.30–130.26	103.07 ± 12.54	75.69–130.18	0.875	136.00 ± 21.01	81.98–190.01	94.66 ± 14.12	58.35–130.97	0.172
**Glucose (mg/dL)**	88.85 ± 1.43	85.75–91.96	86.78 ± 1.97	82.52–91.04	0.509	89.16 ± 1.70	84.79–93.54	88.00 ± 2.46	81.66–94.33	0.753
**Total Cholesterol (mg/dL)**	189.42 ± 11.59	164.37–214.48	191.28 ± 11.57	166.27–216.29	0.944	212.33 ± 13.30	178.12–246.54	211.33 ± 15.64	171.11–251.55	0.600
**HDL Cholesterol (mg/dL)**	70.28 ± 4.05	61.53–79.04	76.92 ± 5.39	65.26–88.59	0.035 *	73.83 ± 5.98	58.44–89.21	72.83 ± 6.89	55.10–90.56	0.892
**LDL Cholesterol (mg/dL)**	113.35 ± 9.55	92.72–133.99	108.42 ± 8.59	89.85–126.99	0.421	131.83 ± 13.85	96.21–167.45	137.00 ± 17.03	93.21–180.78	0.685
**Triglycerides (mg/dL)**	74.92 ± 7.06	59.67–90.18	67.85 ± 7.21	52.27–83.43	0.330	91.50 ± 8.95	68.47–114.52	67.66 ± 15.90	26.77–108.56	0.172

**Table 2 ijerph-19-09363-t002:** Comparison of blood biochemical parameters in men from the probiotic and placebo groups at the initial and final stages of the study. * Statistical significance (*p* < 0.05) determined between the initial and final stages.

Probiotic Men (*n* = 20)						Placebo Men (*n* = 26)				
	Initial Stage		Final Stage			Initial Stage		Final Stage		
Parameter	Mean ± SEM	95% CI	Mean ± SEM	95% CI	*p*	Mean ± SEM	95% CI	Mean ± SEM	95% CI	*p*
**WBC (×10^3^/µL)**	6.12 ± 0.29	5.50–6.73	5.88 ± 0.25	5.34–6.42	0.550	5.63 ± 0.30	5.00–6.26	5.44 ± 0.22	4.97–5.90	0.602
**RBC (×10^6^/µL)**	4.96 ± 0.04	4.86–5.06	4.96 ± 0.06	4.82–5.09	0.896	5.00 ± 0.05	4.88–5.11	5.00 ± 0.06	4.87–5.13	0.919
**HGB (g/dL)**	14.80 ± 0.17	14.43–15.17	14.68 ± 0.18	14.30–15.06	0.344	15.13 ± 0.18	14.75–15.52	14.89 ± 0.21	14.45–15.32	0.061
**PLT (×10^3^/µL)**	238.70 ± 9.60	218.60–258.79	234.95 ± 8.52	217.09–252.80	0.717	230.88 ± 10.47	209.31–252.45	234.07 ± 9.75	213.99–254.15	0.572
**Sodium (mmol/L)**	138.50 ± 0.39	138.12–139.77	139.30 ± 0.61	138.01–140.58	0.426	138.00 ± 0.23	137.51–138.48	138.84 ± 0.31	138.20–139.48	0.061
**Potassium (mmol/L)**	4.76 ± 0.10	4.53–4.99	4.83 ± 0.13	4.56–5.11	0.837	4.63 ± 0.06	4.51–4.76	4.62 ± 0.07	4.47–4.76	0.969
**Calcium (mmol/L)**	2.48 ± 0.02	2.43–2.52	2.42 ± 0.02	2.38–2.47	0.011 *	2.43 ± 0.01	2.39–2.47	2.42 ± 0.01	2.38–2.46	0.786
**Magnesium (mmol/L)**	0.87 ± 0.01	0.83–0.91	0.83 ± 0.01	0.80–0.86	0.211	0.81 ± 0.01	0.79–0.83	0.83 ± 0.00	0.81–0.85	0.034 *
**Iron (µg/dL)**	109.55 ± 11.70	85.06–134.03	105.50 ± 8.45	87.80–123.19	0.866	111.69 ± 9.09	92.95–130.43	112.96 ± 10.36	91.60–134.31	0.590
**Glucose (mg/dL)**	98.85 ± 2.03	94.58–103.11	98.00 ± 2.93	91.85–104.14	0.235	90.34 ± 0.94	88.39–92.29	91.65 ± 1.20	89.16–94.13	0.294
**Total Cholesterol (mg/dL)**	195.90 ± 10.34	174.25–217.54	200.05 ± 9.62	179.90–220.19	0.422	190.15 ± 8.82	171.98–208.32	189.23 ± 7.55	173.67–204.78	0.969
**HDL Cholesterol (mg/dL)**	54.35 ± 3.35	47.32–61.37	52.80 ± 2.74	47.05–58.54	0.375	60.19 ± 2.88	54.24–66.13	60.11 ± 2.56	54.82–65.40	0.931
**LDL Cholesterol (mg/dL)**	134.25 ± 10.19	112.91–155.58	146.25 ± 11.25	122.69–169.80	0.038 *	120.34 ± 8.10	103.65–137.03	128.46 ± 8.46	111.03–145.88	0.137
**Triglycerides (mg/dL)**	107.95 ± 11.09	84.73–131.16	119.95 ± 21.47	75.00–164.89	0.935	86.80 ± 8.45	69.39–104.22	79.73 ± 6.45	66.43–93.00	0.858

**Table 3 ijerph-19-09363-t003:** Comparison of the diet of men and women from the probiotic and placebo groups. * Statistical significance (*p* < 0.05) determined between the initial and final stages.

Women (*n* = 20)						Men (*n* = 46)				
	Probiotic (*n* = 14)		Placebo (*n* = 6)			Probiotic (*n* = 20)		Placebo (*n* = 26)		
Parameter	Mean ± SEM	95% CI	Mean ± SEM	95% CI	*p*	Mean ± SEM	95% CI	Mean ± SEM	95% CI	*p*
Energy (kcal)	1716.45 ± 108.18	1482.73–1950.15	1321.90 ± 65.32	1153.98–1489.82	0.018 *	1926.55 ± 95.66	1726.31–2126.77	2192.81 ± 152.49	1878.74–2506.88	0.369
Protein (g)	79.79 ± 5.75	67.36–92.22	70.20 ± 8.44	48.50–91.90	0.386	86.82 ± 4.74	76.89–96.74	107.71 ± 7.79	91.66–123.76	0.055
Animal Protein (g)	52.21 ± 6.14	38.94–65.48	46.33 ± 9.53	21.82–70.83	0.710	52.59 ± 4.58	42.99–62.18	70.82 ± 7.45	55.47–86.17	0.024 *
Vegetable Protein (g)	27.28 ± 2.45	21.97–32.58	21.67 ± 1.20	18.56–24.77	0.302	32.74 ± 2.77	26.93–38.53	34.82 ± 2.83	28.98–40.65	0.665
Fat (g)	61.34 ± 6.70	46.84–75.83	46.43 ± 7.19	27.94–64.91	0.201	68.35 ± 4.90	58.09–78.61	80.62 ± 9.10	61.86–99.38	0.798
Saturated Fatty Acids (g)	21.59 ± 2.46	16.25–26.92	15.31 ± 2.41	9.09–21.52	0.231	23.67 ± 1.87	19.75–27.58	27.10 ± 3.20	20.50–33.69	0.991
Monounsaturated Fatty Acids (g)	23.93 ± 3.19	17.03–30.82	17.46 ± 2.40	11.28–23.62	0.302	26.93 ± 1.81	23.14–30.73	34.12 ± 4.30	25.26–42.99	0.833
Polyunsaturated Fatty Acids (g)	10.25 ± 1.35	7.32–13.18	8.73 ± 2.27	2.89–14.57	0.536	12.12 ± 1.53	8.91–15.33	12.68 ± 1.33	9.93–15.43	0.850
Cholesterol (mg)	295.23 ± 43.72	200.77–389.69	350.23 ± 46.94	229.55–470.91	0.386	325.28 ± 39.18	243.26–407.30	517.89 ± 71.02	371.61–664.17	0.023 *
Carbohydrates (g)	225.26 ± 13.77	195.49–255.02	173.13 ± 10.68	145.67–200.59	0.028 *	259.13 ± 14.42	228.93–289.32	270.14 ± 19.06	230.87–309.42	0.991
Fiber (g)	21.03 ± 1.84	17.06–25.00	19.88 ± 1.85	15.10–24.65	0.710	24.12 ± 1.83	20.28–27.97	24.62 ± 2.00	20.48–28.76	0.991
Sodium (mg)	2916.02 ± 203.36	2476.68–3355.35	2207.53 ± 209.76	1668.31–2746.74	0.043 *	3330.74 ± 221.26	2867.63–3793.85	3919.91 ± 264.89	3374.34–4465.48	0.180
Potassium (mg)	3461.72 ± 214.36	2998.62–3924.82	2866.12 ± 326.74	2026.19–3706.05	0.173	3812.51 ± 280.17	3226.09–4398.91	3644.05 ± 162.00	3310.38–3977.71	0.850
Calcium (mg)	723.49 ± 54.53	605.67–841.30	555.04 ± 54.14	415.86–694.21	0.063	767.56 ± 79.23	601.71–933.39	771.57 ± 70.26	626.85–916.30	0.991
Magnesium (mg)	373.88 ± 25.77	318.20–429.55	313.24 ± 27.96	241.36–385.11	0.201	391.20 ± 23.89	341.18–441.20	414.74 ± 23.78	365.75–463.73	0.587
Iron (mg)	12.31 ± 1.05	10.04–14.58	10.86 ± 1.05	8.15–13.57	0.386	13.96 ± 0.83	12.21–15.70	15.39 ± 0.90	13.52–17.26	0.324
Vitamin B_6_ (mg)	1.83 ± 0.13	1.55–2.11	1.73 ± 0.29	0.95–2.49	0.901	2.07 ± 0.14	1.77–2.37	2.18 ± 0.11	1.94–2.41	0.444
Vitamin B_12_ (μg)	4.43 ± 1.16	1.92–6.94	3.02 ± 0.35	2.09–3.94	0.967	4.76 ± 0.92	2.81–6.70	5.30 ± 0.59	4.08–6.53	0.134
Vitamin C (mg)	98.41 ± 10.82	75.03–121.78	165.93 ± 50.02	37.35–294.51	0.231	96.83 ± 12.74	70.16–123.49	101.27 ± 9.19	82.34–120.20	0.324
Vitamin D (μg)	3.47 ± 0.91	1.51–5.44	2.15 ± 0.34	1.27–3.03	0.967	3.91 ± 0.71	2.41–5.42	4.97 ± 0.63	3.66–6.27	0.227

## Data Availability

Data available on request due to restrictions e.g., privacy or ethical.

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
