# Peer review of "Effect of a Multi-Strain Probiotic Supplement on Gastrointestinal Symptoms and Serum Biochemical Parameters of Long-Distance Runners: A Randomized Controlled Trial"

_ijerph, 2022, doi:10.3390/ijerph19159363_

Round 1

Reviewer 1 Report

The manuscript ijerph-1847848  entitled “Effect of a multi-strain probiotic supplementation on gastrointestinal symptoms and serum biochemical parameters of long-distance runners: a randomized controlled trial” was allowed to re-summitted to International Journal of Environmental Research and Public Health. I as one of the previous reviewers considered that this manuscript is of particular interest. The authors tried to do best to response most of my comments and suggestions. In addition, the authors had this manuscript more clearly than the previous submitted manuscript. They described how to random volunteer which means they have reliable criteria and experimental design to set up the experiments. Here, I found some minor errors in the manuscripts as follows.

1. Line 1 please considered changing “practicing” to “practice”

2. Line 69, 371, and 372 please do not italicize “subsp.”

3. Line 127 please change “L. lactis” to “Lc. lactis” because L. is the abbreviation of Lactobacillus.

4. Line 364 please change “L. fermentum” to “Limosilactobacillus fermentum”.

5. Line 366 please change “L. fermentum” to “Ls. fermentum”.

6. Line 375 please change “L. rhamnosus” to “Limosilactobacillus rhamnosus”.

7. Line 394 please change “L. plantarum” to “Lactiplantibacillus plantarum”

8. Line 407 please change “L. plantarum” to “Lp. plantarum”

9. References 11, 28, 38, 42, 43, and 45, journal name was abbreviated while others were present in full name. Please once again check reference style of ijerph.

10. Table 1 at p =0.035 (HDL), Table 2 at p=0.011 (calcium), p=0.038 (LDL), please reconsider and/or recalculate these values again due to values between initial and final stages are quite similar.

Author Response

The manuscript ijerph-1847848  entitled “Effect of a multi-strain probiotic supplementation on gastrointestinal symptoms and serum biochemical parameters of long-distance runners: a randomized controlled trial” was allowed to re-summitted to International Journal of Environmental Research and Public Health. I as one of the previous reviewers considered that this manuscript is of particular interest. The authors tried to do best to response most of my comments and suggestions. In addition, the authors had this manuscript more clearly than the previous submitted manuscript. They described how to random volunteer which means they have reliable criteria and experimental design to set up the experiments. Here, I found some minor errors in the manuscripts as follows.

Thank you very much for review your manuscript and kind comments. We tried very hard to make our manuscript as good as possible. Thank you very much for any comments, our answers below.

  1. Line 1 please considered changing “practicing” to “practice”

Thank you, we changed “practicing” to “practice”.

  1. Line 69, 371, and 372 please do not italicize “subsp.”

Thank you, changes was made in the manuscript.

  1. Line 127 please change “L. lactis” to “Lc. lactis” because L. is the abbreviation of Lactobacillus.

Thank you, changes was made in the manuscript.

  1. Line 364 please change “L. fermentum” to “Limosilactobacillus fermentum”.

Thank you, changes was made in the manuscript.

  1. Line 366 please change “L. fermentum” to “Ls. fermentum”.

Thank you, changes was made in the manuscript.

  1. Line 375 please change “L. rhamnosus” to “Limosilactobacillus rhamnosus”.

Thank you, changes was made in the manuscript.

  1. Line 394 please change “L. plantarum” to “Lactiplantibacillus plantarum”

Thank you, changes was made in the manuscript.

  1. Line 407 please change “L. plantarum” to “Lp. plantarum”

Thank you, changes was made in the manuscript.

  1. References 11, 28, 38, 42, 43, and 45, journal name was abbreviated while others were present in full name. Please once again check reference style of ijerph.

Thank you, changes was made in the manuscript.

  1. Table 1 at p =0.035 (HDL), Table 2 at p=0.011 (calcium), p=0.038 (LDL), please reconsider and/or recalculate these values again due to values between initial and final stages are quite similar.

Thank you very much, it is an important comment for us. We re-assessed the statistical significance and obtained the same results as in the manuscript.

This manuscript is a resubmission of an earlier submission. The following is a list of the peer review reports and author responses from that submission.

Round 1

Reviewer 1 Report

The study has some serious flaws:

1)     Methodology

a.      Why there are such a different number of athletes in the intervention vs placebo group? 14 vs 6 in women…..

b.      Explain better the level of the athletes and the groups studied (days per week, type of training……)

c.      Explain the procedures to assign to one group or another

d.      Indicate the manufacturer and the country of the company of computer program Dieta 5. Has it been validated?

e.      Indicate the manufacturer and the country of the company for statistical program STATISTICA 13.3 issued by Stat Soft.

2)     Results

a.      Include d effects analysis to allocate and corroborate the effect size. How can you be sure the probiotics are the responsible for such a weak significance?

b.      Remove the median from the table and include 95%CI

3)     Discussion

a.      Include limitations of the study. (It has many)

Introduction: indicate the level of the athletes and the groups studied

Minor spelling mistakes. A deep review is needed by a native speaker. Some are indicated here:

-        Line 15: a for an

-        Line 44: “one of the theories…..”

-        Line 137: included instead of include

-        Line 281: Rewrite for a better understanding

Author Response

Dear Sir/Madam,

Thank you very much for your consideration of this manuscript and for your review. Below are the answers to the questions.

The study has some serious flaws:

1)   Methodology

  1. Why there are such a different number of athletes in the intervention vs placebo group? 14 vs 6 in women…..

      Thank you for this comment. While recruiting the participants to the study, mainly men applied for the research, and the number of women running very long distances was limited. The authors of the study tried very hard to include both men and women in the study, however, due to the inclusion and exclusion criteria, it was very hard. Moreover, during the study, 4 people (women) resigned from the study (not giving the reason or due to a knee joint injury) - what we mentioned in the manuscript. The number of women in the groups was also influenced by their allocation to groups, which we will describe in detail in point c.

  1. Explain better the level of the athletes and the groups studied (days per week, type of training……)

      Thank you for this comment. The study involved long-distance runners, actively participating in sports competitions with distances >100 km. The runners made endurance and long-distance efforts (>5 km run per day; 5-7 day per week). In addition, the runners did strengthening training 1-2 times a week for about 45 minutes each. The average duration of run training per week in the PRO group was 4.6 h for women and 8.4 h for men, while in the PLA group it was on average 8.3 h for women and 6.7 h for men. Changes have been made to the manuscript (for better understanding by the reader).

  1. Explain the procedures to assign to one group or another.

      Thank you for this comment. The participants who apply for the study were immediately assigned to the PRO or PLA group (due to the fact that it was a double-blind, placebo-controlled trial, the researchers used the designations of groups A and B, only in the manuscript it is described as PRO and PLA). Runners were divided according to simple random sampling without replacement. Every second person drawn was the group A (PRO (n=35 people)), and the rest of the people - group B (PLA (n=35 people)). The order of people reporting for the study caused differences in the number of women in the study group and the control group. There are no such large differences among men due to more frequent application for research. Changes have been made to the manuscript (for better understanding by the reader).

  1. Indicate the manufacturer and the country of the company of computer program Dieta 5. Has it been validated?

      “Dieta 5” program was created by the Polish Institute of Food and Nutrition / National Institute of Public Health, Warsaw, Poland. Changes have been made to the manuscript. The authors of the Diet 5 program were asked if the program was validated, but no response was obtained when preparing the response to the review. Below article in which the “Dieta 5” program was also used:

  1. Małgorzata Szczuko, Rafał Romaniuk, Low salicylate diet and the possibility of nutrient deficiencies, Pomeranian J Life Sci, 2016;62(4):18-24.
  2. Indicate the manufacturer and the country of the company for statistical program STATISTICA 13.3 issued by StatSoft.

      “STATISTICA 13.3” program is created by the StatSoft, Cracow, Poland. Changes have been made to the manuscript.

2) Results

  1. Included effects analysis to allocate and corroborate the effect size. How can you be sure the probiotics are the responsible for such a weak significance?

      We tried to do the most careful and reliable research. Runners were divided according to simple random sampling without replacement. Certainly, our goal for the future is to increase the research group (it is very difficult in sport) so that these conclusions can be even more confirmed.

  1. Remove the median from the table and include 95%CI

     Changes have been made to the manuscript.

3)  Discussion

  1.  Include limitations of the study. (It has many)

     Study limitations are in the manuscript (last paragraph of discussion). It was very important to us to present all the limitations of the work, to show other authors what we believe to do better after our research.

     “The study has some limitations. One of them is a small sample size (n=66), due to the high cost of the analyses. It is known that studies with larger sample sizes are more representative and more reliable. Furthermore, since there are more male than female runners, the genders may not be equally represented.  Additionally, a study of a longer duration may have allowed for more precise conclusions to be drawn. Another factor limiting our study is the diversity in the microbiome of study participants, which would require host-specific selection of probiotic strains. Research investigating the most appropriate probiotic strains for in reducing GI symptoms in athletes is still ongoing. Furthermore, the same probiotic was given to all participants from the probiotic groups during the trial, although each individual reported different GI problems at the beginning of the study were different.

Introduction: indicate the level of the athletes and the groups studied

Thank you. In the introduction section, information on the intensity of exercise that influences the occurrence of gastrointestinal symptoms has been added. In addition, the M&M section thoroughly show the intensity and type of physical effort undertaken by the participants.

Minor spelling mistakes. A deep review is needed by a native speaker. Some are indicated here:

-        Line 15: a for an

-        Line 44: “one of the theories…..”

-        Line 137: included instead of include

-        Line 281: Rewrite for a better understanding

The linguistic proofreading was done by a professional native speaker and the corrections are included in the manuscript.

Reviewer 2 Report

Overall, the manuscript ijerph-1785350 entitled “multi-strain probiotic supplementation – does it effect on gastrointestinal disorder and serum biochemical parameters of long-distance runners? A randomized controlled trail.” is of particular interest due to large numbers of evaluated volunteers which I concerned their difficulties. The results demonstrated that intake of multi-strain probiotics could improve gastrointestinal disorders of men and women which the authors evaluated the effect of probiotic by completing questionnaires. Intake of probiotic seems not to improve blood biochemical parameters for men but women. However, it is fine to me and readers to present this data since the the data obtained from large number of volunteers and there may be some limitations and unavoidable variations. After the evaluation, I recommend accepting this manuscript with some conditions due to mistakes and unclear information as follows.

1. Please avoid initiating sentence with number. For example line 28, “70% of athletes …..”

2. Line 58-61: I would be nice to explain how intense physical exertion adversely affected the intestinal barrier.

3.  Line 94: How did the authors defined moderate or intense physical activity?

4. Line 196-200: Please indicate number of people for 57%  women (n=??) and 40% men (n=??)

5. Line 244-245: Please make the sentence clearly, the small differences were not statistically significant.

6. Table 1 to Table 4, I suggest removing the star symbol at p*. Please add the symbol to where the p-value is of significance. Also, please indicate the significant level e.g., p<0.05 or whatever.

7. When I looked at the significant values such as HDL cholesterol in Table 1, 70.28±15.16 and 76.92±20.20. These values are apparently not significant. I understand that most publication generally present data with mean value together with standard deviation but in fact statistical analysis considers mean values with “standard error”. It would be clear to me and readers if the authors could edit and present their data by mean ± standard error.

8. Line 293-294: Is there any values indicated that there were no statistically significant differences in the diet after 3 months of research? In Table 3, I found only some significant differences in diet parameters such as energy, carbohydrate in group of women.

9.  Please avoid using “own” in the manuscript.

10. Table 3, the table will be very informative if there are recommended standard for each diet parameter.

12. Line 401-404: I do not agree with the sentence (flavorable improvement) since there were no significant differences in sodium, potassium and iron concentrations in group of women. Please modify or edit the sentence.
13. Line 420-430: Please remove line 420-430 ( and the studied men…… in this group of runners.).

14. Line 431-445: Please give reasons why probiotic showed positive effect only in women, not men.

15. Please edit new name of some probiotic bacteria according to an update of bacterial classification.

Lactobacillus brevis  à Levilactobacillus brevis

Lactobacillus salivarius à Ligilactobacillus salivarius

The full name of microorganisms should be written only for the first time, after that the abbreviated their genus. For example, Lactobacillus acidophilus – L. acidophilus; Levilactobacillus brevis – Lv. brevis; Ligilactobacillus salivarius – Lg. salivarius. Please carefully check throughout the manuscript.

16. Line 19-20: As I did not see any data showing a trend of increasing serum iron concentration as stated in abstract. Please clarify

17. Line 22-23: Please clarify “but should be selected individually to the needs of the competitors.” If it is no significant, please consider removing it.

18. Please check the reference style. There are errors, specifically Journal name and abbreviation. Scientific name of microorganisms should be italicized. 

Author Response

Dear Sir/Madam,

Thank you very much for your consideration of this manuscript and for your review. Below are the answers to the questions.

Overall, the manuscript ijerph-1785350 entitled “multi-strain probiotic supplementation – does it effect on gastrointestinal disorder and serum biochemical parameters of long-distance runners? A randomized controlled trail.” is of particular interest due to large numbers of evaluated volunteers which I concerned their difficulties. The results demonstrated that intake of multi-strain probiotics could improve gastrointestinal disorders of men and women which the authors evaluated the effect of probiotic by completing questionnaires. Intake of probiotic seems not to improve blood biochemical parameters for men but women. However, it is fine to me and readers to present this data since the the data obtained from large number of volunteers and there may be some limitations and unavoidable variations. After the evaluation, I recommend accepting this manuscript with some conditions due to mistakes and unclear information as follows.

Thank you very much for that kindly comments.

  1. Please avoid initiating sentence with number. For example line 28, “70% of athletes …..”

Thank you for this comment. The linguistic proofreading was done by a professional native speaker and the corrections are included in the manuscript.

  1. Line 58-61: I would be nice to explain how intense physical exertion adversely affected the intestinal barrier.

      Thank you for this comment. The intestinal barrier may be adversely affected by very intense (≥60–70% VO2max) physical exertion (circulatory-gastrointestinal pathway (redistributes blood flow to working muscles and peripheral circulation, reducing total splanchnic perfusion) and neuroendocrine pathway (the increase in sympathetic activation and the consequent reduction in the gastrointestinal functional capacity)), taking medications, especially non-steroidal anti-inflammatory drugs (NSAIDs) and proton pump inhibitors (PPIs), as well as chronic stress (also associated with participation in competitions). Changes have been made to the manuscript and additional reference from 2021 was added.

  1. Line 94: How did the authors defined moderate or intense physical activity?

According to the ACSM (SWAIN, DAVID P.; ABERNATHY, KIMBERLY S.; SMITH, CARLA S.; LEE, SHIRLEY J.; BUNN, SHELLY A. Target heart rates for the development of cardiorespiratory fitness, Medicine & Science in Sports & Exercise: January 1994 - Volume 26 - Issue 1 - p 112-116) suggestion we defined moderate physical activity: 60% VO2max which corresponds to 70% HRmax, and intense physical activity: 80% VO2max corresponds to 85% HRmax. For the age group of the competitors it was determined that the heart rate for moderate exercise should be around 112-140 bpm / minute, and for intense exercise: 136-170 bpm / minute. The participants were asked about it before starting the study. Changes have been made to the manuscript.

  1. Line 196-200: Please indicate number of people for 57% women (n=??) and 40% men (n=??).

Thank you for this comment. Changes have been made to the manuscript.

  1. Line 244-245: Please make the sentence clearly, the small differences were not statistically significant.

Thank you for this comment. Changes have been made to the manuscript.

  1. Table 1 to Table 4, I suggest removing the star symbol at p*. Please add the symbol to where the p-value is of significance. Also, please indicate the significant level e.g., p<0.05 or whatever.

Thank you for this comment. Changes have been made to the manuscript.

  1. When I looked at the significant values such as HDL cholesterol in Table 1, 70.28±15.16 and 76.92±20.20. These values are apparently not significant. I understand that most publication generally present data with mean value together with standard deviation but in fact statistical analysis considers mean values with “standard error”. It would be clear to me and readers if the authors could edit and present their data by mean ± standard error.

Thank you for this comment. Changes have been made to the manuscript.

  1. Line 293-294: Is there any values indicated that there were no statistically significant differences in the diet after 3 months of research? In Table 3, I found only some significant differences in diet parameters such as energy, carbohydrate in group of women.

The competitors were asked that their diet did not change throughout the research process (3 months) - if any changes occurred, the researchers were to be informed about it. After collecting the data on the diet after 3 months - there were no statistically significant differences in it, as we wrote “It was demonstrated that the lack of changes in the diet indicates that no influence of the intake of energy and selected nutrients, minerals and vitamins on the incidence of GI symptoms”. If the reviewer think it is necessary - we can prepare a table with the results of the diet after 3 months (we have not done it so far because the manuscript came out long and we did not want to additionally enter irrelevant data in it).

  1. Please avoid using “own” in the manuscript.

Thank you for this comment. The linguistic proofreading was done by a professional native speaker and the corrections are included in the manuscript.

  1. Table 3, the table will be very informative if there are recommended standard for each diet parameter.

We were not able to modify the table in the manuscript - too many comments and changes, so we attach the table below. It is very difficult to establish norms for these runners, the authors focused on scientific publications on sport and norms for the Polish population. We tried to select the norm for the given ingredients as precisely as possible so that the reader had the greatest transparency of the result.

Table 3. Comparison of the diet of men and women from the probiotic and placebo groups. * Statistical significance (p<0.05) determined between the initial and final stages. Standards for individual components have been prepared based on [14-30].

Women (n=20)

Men (n=46)

PRO (n=14)

PLA (n=6)

PRO (n=20)

PLA (n=26)

Parameter

Norm for women

Mean±SEM

95% CI

Mean±SEM

95% CI

p

Norm for men

Mean±SEM

95%CI

Mean±SEM

95%CI

p

Energy (kcal)

2500-2600

1716.45±108.18

1482.73-1950.15

1321.90±65.32

1153.98-1489.82

0.018*

3200

1926.55±95.66

1726.31-2126.77

2192.81±152.49

1878.74-2506.88

0.369

Protein (g)

95-99

79.79±5.75

67.36-92.22

70.20±8.44

48.50-91.90

0.386

119-121

86.82±4.74

76.89-96.74

107.71±7.79

91.66-123.76

0.055

Animal Protein (g)

64-66

52.21±6.14

38.94-65.48

46.33±9.53

21.82-70.83

0.710

80-81

52.59±4.58

42.99-62.18

70.82±7.45

55.47-86.17

0.024*

Vegetable Protein (g)

31-33

27.28±2.45

21.97-32.58

21.67±1.20

18.56-24.77

0.302

39-40

32.74±2.77

26.93-38.53

34.82±2.83

28.98-40.65

0.665

Fat (g)

95-98

61.34±6.70

46.84-75.83

46.43±7.19

27.94-64.91

0.201

120-127

68.35±4.90

58.09-78.61

80.62±9.10

61.86-99.38

0.798

Saturated Fatty Acids (g)

<28

21.59±2.46

16.25-26.92

15.31±2.41

9.09-21.52

0.231

<35

23.67±1.87

19.75-27.58

27.10±3.20

20.50-33.69

0.991

Monounsaturated Fatty Acids (g)

42-43

23.93±3.19

17.03-30.82

17.46±2.40

11.28-23.62

0.302

53

26.93±1.81

23.14-30.73

34.12±4.30

25.26-42.99

0.833

Polyunsaturated Fatty Acids (g)

25-27

10.25±1.35

7.32-13.18

8.73±2.27

2.89-14.57

0.536

32-39

12.12±1.53

8.91-15.33

12.68±1.33

9.93-15.43

0.850

Cholesterol (mg)

<300

295.23±43.72

200.77-389.69

350.23±46.94

229.55-470.91

0.386

<300

325.28±39.18

243.26-407.30

517.89±71.02

371.61-664.17.

0.023*

Carbohydrates (g)

315-330

225.26±13.77

195.49-255.02

173.13±10.68

145.67-200.59

0.028*

395-405

259.13±14.42

228.93-289.32

270.14±19.06

230.87-309.42

0.991

Fiber (g)

25-27

21.03±1.84

17.06-25.00

19.88±1.85

15.10-24.65

0.710

25-27

24.12±1.83

20.28-27.97

24.62±2.00

20.48-28.76

0.991

Sodium (mg)

1500

2916.02±203.36

2476.68-3355.35

2207.53±209.76

1668.31-2746.74

0.043*

1500

3330.74±221.26

2867.63-3793.85

3919.91±264.89

3374.34-4465.48

0.180

Potassium (mg)

3500

3461.72±214.36

2998.62-3924.82

2866.12±326.74

2026.19-3706.05

0.173

3500

3812.51±280.17

3226.09-4398.91

3644.05±162.00

3310.38-3977.71

0.850

Calcium (mg)

1000

723.49±54.53

605.67-841.30

555.04±54.14

415.86-694.21

0.063

1000

767.56±79.23

601.71-933.39

771.57±70.26

626.85-916.30

0.991

Magnesium (mg)

320

373.88±25.77

318.20-429.55

313.24±27.96

241.36-385.11

0.201

420

391.20±23.89

341.18-441.20

414.74±23.78

365.75-463.73

0.587

Iron (mg)

18

12.31±1.05

10.04-14.58

10.86±1.05

8.15-13.57

0.386

10

13.96±0.83

12.21-15.70

15.39±0.90

13.52-17.26

0.324

Vitamin B6 (mg)

1.3

1.83±0.13

1.55-2.11

1.73±0.29

0.95-2.49

0.901

1.3

2.07±0.14

1.77-2.37

2.18±0.11

1.94-2.41

0.444

Vitamin B12 (μg)

2.4

4.43±1.16

1.92-6.94

3.02±0.35

2.09-3.94

0.967

2.4

4.76±0.92

2.81-6.70

5.30±0.59

4.08-6.53

0.134

Vitamin C (mg)

75

98.41±10.82

75.03-121.78

165.93±50.02

37.35-294.51

0.231

90

96.83±12.74

70.16-123.49

101.27±9.19

82.34-120.20

0.324

Vitamin D (μg)

15

3.47±0.91

1.51-5.44

2.15±0.34

1.27-3.03

0.967

15

3.91±0.71

2.41-5.42

4.97±0.63

3.66-6.27

0.227

  1. Line 401-404: I do not agree with the sentence (flavorable improvement) since there were no significant differences in sodium, potassium and iron concentrations in group of women. Please modify or edit the sentence.

Thank you for this comment. Changes have been made to the manuscript, new sentence: At 3 months after the initiation of the intervention, the concentrations of sodium, potassium and iron in the sera of probiotic women increased, while the concentrations of iron decreased significantly in the placebo women group

  1. Line 420-430: Please remove line 420-430 ( and the studied men…… in this group of runners.).

Thank you for this comment. I wasn't sure if it was about line 420-430 or 429-430 (the words quoted show 429-430) and I deleted words “(…) and the studied men did not supplement this vitamin - this may explain the reduction in serum calcium in this group of runners.”.

  1. Line 431-445: Please give reasons why probiotic showed positive effect only in women, not men.

We cannot show why the probiotic only showed a positive effect in women. In the literature review we have carried out, possible information on the effects of probiotics is presented: cholesterol assimilation by the bacterial cell wall, decreased intestinal cholesterol absorption, increased production of SCFA, which can reduce endogenous cholesterol synthesis. We found no single reports of gender polymorphism. So far, the mechanism of the influence of probiotic strains on cholesterol concentration is unknown.

Vourakis M, Mayer G, Rousseau G. The Role of Gut Microbiota on Cholesterol Metabolism in Atherosclerosis. Int J Mol Sci. 2021 Jul 28;22(15):8074

Olas B. Probiotics, Prebiotics and Synbiotics-A Promising Strategy in Prevention and Treatment of Cardiovascular Diseases? Int J Mol Sci. 2020 Dec 20;21(24):9737.

  1. Please edit new name of some probiotic bacteria according to an update of bacterial classification. Lactobacillus brevis à Levilactobacillus brevis, Lactobacillus salivarius à Ligilactobacillus salivarius

Thank you very much for this very important comment! The changes have been made in the manuscript.

The full name of microorganisms should be written only for the first time, after that the abbreviated their genus. For example, Lactobacillus acidophilus – L. acidophilus; Levilactobacillus brevis – Lv. brevis; Ligilactobacillus salivarius – Lg. salivarius. Please carefully check throughout the manuscript.

Thank you very much for this comment! The changes have been made in the manuscript.

  1. Line 19-20: As I did not see any data showing a trend of increasing serum iron concentration as stated in abstract. Please clarify

Thank you very much for this comment. Upon reflection - this sentence could be misleading. An increase in serum iron concentration in women from the PRO group was noticed after 3 months and a decrease in iron concentration in women from the PLA group - that is why the authors wrote about the possible beneficial effect of the probiotic. Sentence in line 19-20 from abstract was deleted.

  1. Line 22-23: Please clarify “but should be selected individually to the needs of the competitors.” If it is no significant, please consider removing it.

Authors wanted to present with this sentence information about the necessity of individual selection of probiotics to the needs of runners - because the symptoms often differ individually. After discussion, we removed this sentence.

  1. Please check the reference style. There are errors, specifically Journal name and abbreviation. Scientific name of microorganisms should be italicized.

Thank you very much for this comment – the reference style was changed.

Round 2

Reviewer 1 Report

Most of the suggestions were NOT attended / fixed.

Results and introduction are even worse than before

1) Such a different number of athletes in the intervention vs placebo group? 14 vs 6 in women

2) Explain the procedures to assign to one group or another (computer based?.......)

3) the manufacturer and the country of the company must be between brackets

4) Major spelling mistakes in the intro

5) Tables format impossible to read
